# Artificial Intelligent Life: A New Perspective on Artificial General Intelligence

**Borui Cai, Yong Xiang, Yao Zhao**
School of Information Technology, Deakin University, Australia
`{b.cai,yong.xiang,cnr}@deakin.edu.au`

## Abstract

We propose artificial intelligent life (AILife) as a new perspective to approach artificial general intelligence, similar to a living organism in reality. Unlike machine learning approaches that focus on reward functions and mathematical optimizations, AILife seeks to develop an artificial organism that learns by the mechanism of biological neurons. AILife is composed of a biological-like neuron system to learn from interactions with the world, a sensory system to feel the world, and actuators to perform activities. We show a toy example to explain AILife.

## 1    Introduction

Artificial General Intelligence (AGI) Goertzel & Wang (2007) refers to intelligent systems that reason, learn, and adapt to new scenarios in a way comparable to human intelligence. In the last decade, machine learning paradigms, especially deep learning, have been widely regarded as the most promising pathways to AGI, because they have been successful in various tasks. AlphaGo Silver et al. (2016) beats the best human Go players, and ResNet Szegedy et al. (2017) surpasses human vision for image recognition. But they still face many critical problems to reach AGI, such as lacking generalization for multi-tasks Ruder (2017) and rare/unseen scenarios Ribeiro et al. (2016).

To date, human intelligence still is the only general intelligence. It is developed during interactions with the real world. Infants perceive space and time by instinctively exploring the world through watching, touching, and crawling Casasanto et al. (2010), and gradually develop more complicated capabilities. Many attempts at machine learning have been proposed to mimic such a learning process, e.g, reinforcement learning Matsuo et al. (2022) and autonomous intelligent agent LeCun (2022). However, they cannot perceive space and time as humans do, and are fundamentally different from human intelligence. Specifically, to solve a task, machine learning methods design mathematical models for optimization, while living organisms (including human) learn through complex communications among biological neurons, which are triggered by interacting with the world Shine et al. (2019). Thanks to the advances in neuroscience, the basic mechanism of neuron communication within living organisms has been revealed Hawk et al. (2018), and that provides another pathway (without mathematical reward functions and optimizations) to approach intelligence.

Inspired by that, we propose a new perspective to approach AGI, artificial intelligent life (AILife), aiming at developing intelligence similar to living organisms. To achieve that, AILife involves two key elements: 1) the interaction, AILife lives in a virtual/physical world and interacts with the world following pre-defined activities (instincts), and 2) the learning mechanism, AILife learns by neuron communications triggered by interactions, with biological-like artificial neuron systems.

## 2    Artificial Intelligent Life

AILife aims at developing intelligence by mimicking a living organism in reality. That is, an AILife instance interacts with the virtual/real world based on pre-defined activities, and such interactions trigger neuron communications within biological-like neuron systems of AILife to develop intelligence. To do that, an AILife instance includes an artificial neuron system to learn from interactions by the mechanism of neuron communication, a sensory system to perceive the world, and actuators to perform activities. The sensory signals and actuator actions form the input-to-output circle for the neuron system to learn, similar to the conditioned stimulus in living organisms.

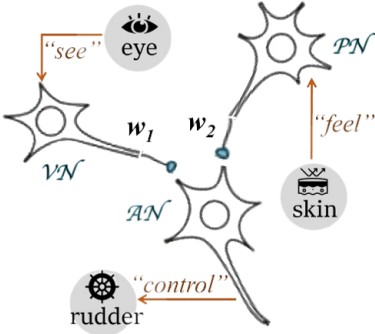

**Figure 1: The neuron system of $\mathcal{P}$.**

We provide a toy example (*smart pinball*) to explain AILife (code is available on Github[1]). We define a virtual one-dimensional world, which has two barriers. A pinball lives between barriers, and its pre-defined activity (instinct) is to constantly move with an invariant speed. The physical rule of this world is that barriers cannot be crossed, and thus the pinball has to change direction when hits them. We expect the pinball to be *smart* to realize this rule by interacting with the barriers, so that it changes direction before hitting them.

To do that, we design *smart pinball* ($\mathcal{P}$) based on the principle of AILife. Specifically, $\mathcal{P}$ contains: 1) a neuron system implemented by spiking neural networks (SNN) Nunes et al. (2022), which is capable of mimicking the biological learning process Taherkhani et al. (2020); 2) a sensory system that includes an "*eye*" to see a short distance in the front and a tactile "*skin*" to feel the hit; and 3) an actuator ("*rudder*") to change direction. By convention, we assume the neuron system with fixed neurons and adjustable connections Wang et al. (2020). A neuron in SNN has inputs to receive signals from other neurons and outputs to fire signals. The signals are transmitted as discrete spikes. When a neuron receives a spike from another neuron, its *membrane potential* increases, based on the weight of the connection. The neuron fires when the *membrane potential* reaches a threshold. After firing, the *membrane potential* returns to the initial value. The learning of SNN is based on Spike-Timing Dependent Plasticity (STDP) Shouval et al. (2010), which mimics the strengthen/weaken mechanism of connections among biological neurons.

The neuron system of $\mathcal{P}$ (Fig. 1) contains three neurons: 1) a vision neuron ($\mathcal{VN}$) that fires when the "*eye*" sees a barrier; 2) a pain neuron ($\mathcal{PN}$) that fires when the "*skin*" feels hitting a barrier; and 3) an action neuron ($\mathcal{AN}$) that fires with threshold $\delta$ to control the "*rudder*" to change direction. The connection of the three neurons are $\mathcal{VN} \rightarrow \mathcal{AN}$ (with weight $w_1$) and $\mathcal{PN} \rightarrow \mathcal{AN}$ (with weight $w_2$). We set the initial $w_1 \ll \delta$ assuming there is no causal relationship between vision and action in the beginning, and the initial $w_2 > \delta$ as feeling pain naturally drives $\mathcal{P}$ to change direction.

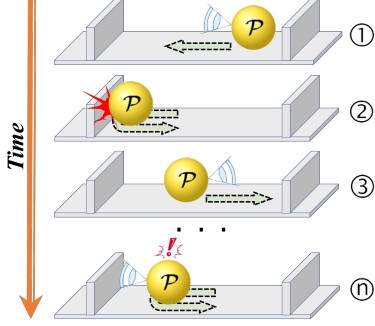

**Figure 2: The learning process of $\mathcal{P}$.**

The learning process of $\mathcal{P}$ is shown in Fig. 2. Initially, $\mathcal{P}$ does not know barriers cannot be crossed. Thus $\mathcal{P}$ keeps moving in one direction (Fig. 2 ①) till it hits a barrier. The hit causes $\mathcal{PN}$ to fire, which further causes $\mathcal{AN}$ to fire, due to $\mathcal{PN} \rightarrow \mathcal{AN}$ with large $w_2$; and that makes $\mathcal{P}$ change the moving direction (Fig. 2 ② and ③). The key point is that, at the same time, $\mathcal{VN}$ also fires as $\mathcal{P}$ can see the barrier in front. According to STDP, the connection $\mathcal{VN} \rightarrow \mathcal{AN}$ is strengthened ($w_1$ increases) because $\mathcal{VN}$ and $\mathcal{AN}$ express a causal relationship. After a few rounds, when $\mathcal{VN} \rightarrow \mathcal{AN}$ grows strong enough ($w_1 > \delta$), the firing of $\mathcal{VN}$ can cause $\mathcal{AN}$ to fire to change direction (Fig. 2 ⓝ). That means, $\mathcal{P}$ has learned to change direction when it sees a barrier.

## 3 DISCUSSION

In this paper, we propose AILife as a new perspective to approach AGI. As preliminary work, we provide a toy example to explain AILife, expecting to attract interested researchers on exploring many exciting challenges ahead. For example, can AILife develop more complicated intelligence in more complex worlds? can an AILife instance interact with other AILife instances to evolve social activities? can AILife learn from highly abstractive information such as languages to gain high-level cognitive abilities? We envision solving these challenges requires the incorporation of more sophisticated computing technologies and cutting-edge neuron sciences. Meanwhile, we believe the development of AILife also can benefit neuroscience, because it provides a platform that allows for the customized construction of artificial organisms and neuron systems.

---

[1] https://github.com/brcai/SmartPinball

URM STATEMENT

The authors acknowledge that at least one key author of this work meets the URM criteria of ICLR 2023 Tiny Papers Track.

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
