# OpenReview forum: "Artificial Intelligent Life: A New Perspective on Artificial General Intelligence"
_ICLR.cc/2023/TinyPapers — Submitted to Tiny Papers @ ICLR 2023_

### Official Review · Reviewer_qaz1 · 2023-03-20

**Confidence:** 3

**Summary Of Contributions:**

The authors present AILife, a new perspective approach that aims at learning from the interaction with the virtual/real world through: i)a neuron system to mimic the biological learning process; ii) a sensory system that includes an "eye" and "skin"; iii) an actuator to change directions.

**Rating:**

Needs Clarification (NC): a submission which does not meet the reviewing criteria and needs clarification for its described problem or solution

**Strengths And Weaknesses:**

## Strengths:
1. The paper is well written and easy to follow.

## Weaknesses:
1. The sub-images of the neurons in Figure 2 are
grainy.
2. It is difficult to grasp the effectiveness of the approach. Please see the "Suggested Changes" section for more details.
3. Other relevant literature is missing, for example [1]


[1] : "A Path Towards Autonomous Machine Intelligence Version" by Yann LeCun.

**Suggested Changes:**

The authors present a new perspective to approach AGI. Due to the limited space provided by the conference, and the topic of the paper, it is difficult to appreciate the proposed approach. It would be interesting to see a more detailed description and comparison with other visions of AGI, such as [1]. Furthermore, I suggest, where possible, creating a simple simulated version of the proposed approach.

---

> ### Author Response · Authors · 2023-04-30
> **Response to Reviewer qaz1**
>
> Thank you very much for reviewing our paper and pointing out our major drawbacks. We have revised the manuscript based on the comments as follows:
>
> A - The sub-images of the neurons in Figure 2 are grainy.
>
> Q - We have fixed the issue in Figure 2.
>
> A - it is difficult to appreciate the proposed approach
>
> Q - We have greatly improved the clarity of the Introduction section in two aspects. 1) Machine learning methods develop intelligence by designing mathematical models for optimization, while living organisms (including human) learn through complex communications among biological neurons, which are triggered by interacting with the world. 2) AILife develops intelligence in the way of living organisms, that is, **the learning process** (interacting with the world) and **the learning mechanism** (neuron communications among biological-like neuron systems). The reason is that we believe such design can enable AILife to perceive the fundamental elements of the physical world, e.g., space and time, like living organisms, and based on that to develop complex intelligence. In contrast, machine learning methods cannot generalize well partly because they rely on mathematical patterns that do not exist in the physical world.
>
> A - It would be interesting to see a more detailed description and comparison with other visions of AGI, such as [1]
>
> Q - In our opinion, although existing AGI approaches (including [1]) try to mimic the learning process and physical structures of living organisms, **they still focus on developing decision-making mathematical models but not biological-like intelligence**. That is the main difference from our proposal.
>
> A - creating a simple simulated version of the proposed approach.
>
> Q - Thanks for the suggestion, we have uploaded the code and a GIF of SmartPinball on https://github.com/brcai/SmartPinball.

---

### Official Review · Reviewer_ozgM · 2023-03-28

**Confidence:** 3

**Summary Of Contributions:**

The paper introduces a new perspective to approach artificial general intelligence which draws parallels to human intelligence. It propositions a system with three components: (1)  a neuron system to learn from interactions, (2) a sensory system to perceive the world, and (3) actuators to perform activities. This is analogous to the humans and the method of learning is analogous to humans perceive, learn and interact with the surroundings.

**Rating:**

High Potential (HP): a submission which meets the reviewing criteria and has potential to make an impact on the field

**Strengths And Weaknesses:**

Strengths:
- Clearly defined problem statement and motivation for the proposed system.
- Relevant work has been clearly explained and discussed in reference to the proposed work.
- The suggested approach has been clearly explained.

Weakness:
- The authors motivate the idea of using this approach for complex tasks and multitask scenarios. However, the section on AILife does not touch upon that aspect. It would be great to include some motivating discussion on it since those are the most interesting applications.

**Suggested Changes:**

Please see above.

---

> ### Author Response · Authors · 2023-04-30
> **Response to Reviewer ozgM**
>
> Thank you very much for reviewing our paper and for your positive comments.
>
> Q - The authors motivate the idea of using this approach for complex tasks and multitask scenarios. However, the section on AILife does not touch upon that aspect. It would be great to include some motivating discussion on it since those are the most interesting applications.
>
> A - Thanks for the suggestions, we have revised the Introduction and Discussion sections accordingly. Through that, we want to emphasize that the aim of AILife is to develop intelligence similar to living organisms. Therefore, to some extent, AILife for complex tasks and multitask is equivalent to how living organisms (including human) develops such abilities. Based on our understanding, the basis of complex activities for human is the perception of the fundemantal elements of the physical world, e.g., space and time. Machine learning methods cannot generalize well partly because they cannot perceive space and time as human does, but rely on patterns that do not correspond to the physical world. Through the toy example, we show an AILife instance can perceive space and learn to avoid hitting barriers in the way of living organisms, and we will explore whether it can develop multitask abilities when facing more complex worlds in our future work.

---

### Comment · Area_Chair_qqTW · 2023-06-03
**Archival**

This work meets the threshold for archival, contents the URM statement and is deanonymized

---

### Meta-Review · Area_Chair_qqTW · 2023-04-06

**Recommendation:** Invite to revise
**Confidence:** 5

**Metareview:**

This paper introduces a new perspective on the implementation of AGI, named AILife, this framework contains three components: (1) a neural system for interaction, (2) a sensory system for perceiving the environment, (3) an actuator for performing actions.
One reviewer thinks the presentation is clear and could have impact on the field, the other thinks the paper needs clarification and misses related works.
Both reviewers agree that the paper is well written and easy to follow.

After reading the paper, the AC agrees with reviewer qaz1 that the papers need more clarification, now the technical details of the implementation of the system is missing, and cannot be reproduced, also the related works refered by reviewer qaz1 need to be discussed, the AC also thinks the literature on reinforcement learning need to be discussed.


**Summary:**

This paper proposes a possible implementation of AGI, the paper is well written and easy to follow, yet important details on the implementation of the framework is lacking, also important related works are not discussed. Thus more clarification from the authors are needed.

**Reason For Not Giving A Higher Recommendation:**


Important details and highly related works are missing.


**Reason For Not Giving A Lower Recommendation:**

N/A

---

> ### Author Response · Authors · 2023-04-28
> **Response to Reviewer qqTW**
>
> Thank you very much for taking the time to meta-review this paper. We have revised the manuscript to address your concerns as follows:
>
> Q - the paper needs more clarification, important related works are not discussed
>
> A - We make significant changes in the Introduction and Conclusion sections to improve clarity. The aim of these changes is to clarify that the proposed AILife tries to develop intelligence similar to living organisms in both:
>
> * **the learning process**, AILife lives in a virtual/physical world and interacts with the world following pre-defined activities (instincts)
>
> * **the learning mechanism**, AILife learns by neuron communications triggered by interactions, with biological-like artificial neuron systems
>
> As a comparison, reinforcement learning or autonomous intelligent agent only mimics **the learning process** of living organisms (the interaction), but **not the learning mechanism**, because they rely on optimizing mathematical reward functions for learning. In our opinion, the biological learning mechanism provides another pathway (without mathematical optimizations, backpropagations) to approach intelligence, and it has already achieved general intelligence (human intelligence) for many years. So, AILife seeks to approach intelligence like a real living organism does.
>
> Q - the technical details of the implementation of the system is missing, and cannot be reproduced
>
> A - Thanks for pointing this out, we have put the code of the SmartPinball on Github (https://github.com/brcai/SmartPinball). The implementation of this toy example is explained in Section 2. Due to the limited space, we refer details of STDP in related references or our codes.

---

### Decision · Program_Chairs · 2023-04-07

Revision accepted; invite to archive

---

> ### Author Response · Authors · 2023-05-30
> **Opt-in for Achieval**
>
> We wish to opt-in for achieval.